# Usefulness of Clinical Definitions of Influenza for Public Health Surveillance Purposes

**DOI:** 10.3390/v12010095

**Published:** 2020-01-14

**Authors:** Àngela Domínguez, Núria Soldevila, Núria Torner, Ana Martínez, Pere Godoy, Cristina Rius, Mireia Jané

**Affiliations:** 1Departament de Medicina, Universitat de Barcelona, 08036 Barcelona, Spain; angela.dominguez@ub.edu (À.D.); nuriatorner@ub.edu (N.T.); 2CIBER Epidemiología y Salud Pública (CIBERESP), 28029 Madrid, Spain; a.martinez@gencat.cat (A.M.); pere.godoy@gencat.cat (P.G.); crius@aspb.cat (C.R.); mireia.jane@gencat.cat (M.J.); 3Agència de Salut Pública de Catalunya, Generalitat de Catalunya, 08005 Barcelona, Spain; 4Institut de Recerca Biomèdica de Lleida, Universitat de Lleida, 25198 Lleida, Spain; 5Agència de Salut Pública de Barcelona, 08023 Barcelona, Spain

**Keywords:** influenza, sentinel surveillance system, performance assessment, case definition, symptoms, primary healthcare physician

## Abstract

This study investigated the performance of various case definitions and influenza symptoms in a primary healthcare sentinel surveillance system. A retrospective study of the clinical and epidemiological characteristics of the cases reported by a primary healthcare sentinel surveillance network for eleven years in Catalonia was conducted. Crude and adjusted diagnostic odds ratios (aDORs) and 95% confidence intervals (CIs) of the case definitions and symptoms for all weeks and epidemic weeks were estimated. The most predictive case definition for laboratory-confirmed influenza was the World Health Organization (WHO) case definition for ILI in all weeks (aDOR 2.69; 95% CI 2.42–2.99) and epidemic weeks (aDOR 2.20; 95% CI 1.90–2.54). The symptoms that were significant positive predictors for confirmed influenza were fever, cough, myalgia, headache, malaise, and sudden onset. Fever had the highest aDOR in all weeks (4.03; 95% CI 3.38–4.80) and epidemic weeks (2.78; 95% CI 2.21–3.50). All of the case definitions assessed performed better in patients with comorbidities than in those without. The performance of symptoms varied by age groups, with fever being of high value in older people, and cough being of high value in children. In patients with comorbidities, the performance of fever was the highest (aDOR 5.45; 95% CI 3.43–8.66). No differences in the performance of the case definition or symptoms in influenza cases according to virus type were found.

## 1. Introduction

Viral upper respiratory tract infections remain a major cause of morbidity and mortality worldwide, with influenza infections being an important cause [1,2]. Influenza viruses A and B cause annual epidemics and produce 3–5 million cases of severe disease and 290,000 to 650,000 respiratory deaths annually [3]. Because influenza epidemics lead to increased social concern each season and the appearance of a novel influenza A subtype virus can cause a pandemic, and disease surveillance is crucial from a public health perspective [4].

The case definition is a key factor in any standardized system of public health surveillance and, ideally, should be based on a combination of signs and symptoms that characterize the condition of interest [5].

The clinical characteristics of influenza, known as influenza-like illness (ILI), are similar to those caused by other viruses causing acute respiratory infection (ARI), and only laboratory confirmation permits a specific disease diagnosis. Because influenza is very common and, during seasonal epidemics, affects 10%–20% of the unvaccinated population [1], it is not feasible to confirm all suspected cases. Due to the lack of specificity of influenza symptoms [6] and because it is necessary to assess the disease burden, quantitative indicators of ILI or ARI are commonly used.

The World Health Organization (WHO) defines ILI as an acute respiratory illness with a measured temperature of ≥38 °C and a cough, with onset within the last 10 days [7].

The European Center for Disease Prevention and Control (ECDC) defines ILI as the sudden onset of symptoms and at least one of following four systemic symptoms: fever or feverishness, malaise, headache, and myalgia, and at least one of the following three respiratory symptoms: cough, sore throat, and shortness of breath. ARI is defined as sudden symptom onset with at least one of the following four respiratory symptoms: cough, sore throat, shortness of breath, coryza, and the clinician’s judgement that the illness is due to an infection [8].

Any influenza surveillance system aims to reliably assess the influenza epidemic activity each season. The system should provide robust, continuous data in order to monitor the trends of clinically diagnosed ILI or ARI in the population studied and in specific groups at increased risk of complications and death [5,9].

Like any public health surveillance system, the influenza surveillance system must evaluate its own performance in relation to its main purposes [10]. The accuracy of the clinical case definition has traditionally been assessed in terms of sensitivity, specificity, and positive predictive value [11]. More recent studies also include the positive predictive likelihood/negative predictive likelihood ratio, named the diagnostic odds ratio (DOR) [12,13,14].

The objective of this study was to investigate the performance of different case definitions and clinical manifestations of ILI and ARI in a primary healthcare influenza sentinel surveillance system.

## 2. Material and Methods

### 2.1. Study Design

A retrospective study of the clinical and epidemiological characteristics of the cases detected in a primary healthcare sentinel surveillance network for eleven years (2008 to 2018) in Catalonia, a Spanish region with 7.6 million inhabitants, was conducted.

The PIDIRAC (Pla d’Informació de les Infeccions Respiratòries Agudes a Catalunya) sentinel surveillance system [15] is constituted of general practitioners and pediatricians from 40–44 primary care centers distributed throughout Catalonia. The mean number of general practitioners involved during the study period was 32 (ranging from 27 to 35) and the mean number of pediatricians was 26 (ranging from 24 to 28). The mean percentage of the population surveyed was 0.97% (ranging from 0.86 to 1.06).

Each sentinel physician collected weekly nasopharyngeal swabs by simple random selection in up to two individuals who presented symptoms compatible with ARI (with no specified case definition) or ILI according to the European ILI case definition [8], namely: sudden symptom onset as well as at least one of four systemic symptoms (fever >37.8 °C or feverishness, malaise, headache, and myalgia) and at least one of three respiratory symptoms (cough, sore throat, and shortness of breath).

The samples were sent to the core laboratory of the PIDIRAC surveillance system for determination of respiratory viruses (influenza viruses A–C, syncytial respiratory virus, parainfluenza viruses 1–4, adenovirus, coronavirus, rhinovirus, metapneumovirus, and bocavirus) by real-time reverse transcriptase-polymerase chain reaction (RT-PCR). Individuals whose clinical samples showed coinfection and those positive for influenza virus C were excluded.

### 2.2. Data Collection

For each individual, the following variables were collected: reported fever >37.8 °C, cough, malaise, headache, myalgia, sore throat, shortness of breath, sudden onset of symptoms, coryza, age, sex, comorbidities (chronic cardiovascular disease; pulmonary disease, including asthma; liver disease; renal disease; metabolic disorders including diabetes mellitus; obesity, defined as a body mass index >40; and immunodeficiency), and microbiological laboratory results.

The ECDC and WHO case definitions for ILI and the ECDC definition of ARI [7,8] were analyzed. Influenza epidemic weeks were defined each season according to the data provided by PIDIRAC.

### 2.3. Statistical Analysis

To investigate the performance of different case definitions and symptoms, we calculated the sensitivity (Se), specificity (Sp), positive predictive value (PPV), positive likelihood ratio (LR), negative LR, crude and adjusted DOR, and 95% confidence intervals (CI) [16] for all weeks (weeks 1–52/53) and for epidemic weeks (weeks with ILI incidence rate above the set epidemic threshold per each season), comparing the cases in which the influenza virus was detected with cases negative for influenza or positive for other respiratory viruses. The analyses were carried out considering all influenza A and B viruses jointly and influenza A and influenza B viruses separately.

To calculate the crude DOR for each case definition and symptom, logistic regression models with the outcome of laboratory-confirmed influenza were constructed. To calculate the adjusted DOR (aDOR), multivariate logistic regression models adjusted by age, sex, comorbidities, and season were constructed using backward selection. The analysis was performed using the SPSS version 25 statistical package and R version 3.5.0 statistical software.

### 2.4. Ethical Considerations

All of the data used in the analysis were collected during routine public health surveillance activities as part of the legislated mandate of the Health Department of Catalonia, which is officially authorized to receive, treat, and temporarily store personal data in cases of infectious disease [17]. All of the data were fully anonymized. All of the study activities formed part of the public health surveillance and therefore were exempt from institutional board review.

## 3. Results

During the study period, 10,830 samples were collected, of which 438 were excluded because of coinfection, and 25 because they were caused by influenza C virus. Therefore, 10,367 samples were analyzed, of which 3241 (31.3%) were positive for influenza (2035 influenza A only and 1196 influenza B only), 50.3% were female, 51.7% were aged <15 years, and 10.9% presented comorbidities. The most frequent clinical manifestations were fever (86.3%) and cough (80.9%; Table 1).

The WHO definition of ILI showed the highest sensitivity (82%) and the lowest specificity (37%), whereas the ECDC ILI case definition showed the lowest sensitivity (58%) and the highest specificity (52%). The most predictive case definition for laboratory-confirmed influenza was the WHO case definition for ILI, in both all weeks (aDOR 2.69; 95% CI 2.42–2.99) and in epidemic weeks (aDOR 2.20; 95% CI 1.90–2.54). The ECDC acute respiratory infection (ARI) definition had the worst performance (aDOR 1.45; 95% CI 1.33–1.58 for all weeks and 1.36; 95% CI 1.21–1.54 for epidemic weeks). The clinical manifestations that were significant positive predictors of laboratory-confirmed influenza were fever, cough, myalgia, headache, malaise, and a sudden onset of symptoms. Fever had the highest aDOR in all weeks (4.03; 95% CI 3.38–4.80) and in epidemic weeks (2.78; 95% CI 2.21–3.50; Table 2).

The performance of the case definitions and the specific clinical manifestations by age group and the presence of comorbidities for all weeks is shown in Table 3, and for epidemic weeks in Table 4. The best performance of the WHO case definition for ILI corresponded to people aged ≥65 years (aDOR 5.87; 95% CI 3.92–8.80 for all weeks, and 3.45; 95% CI 1.95–6.09 for epidemic weeks). In patients with comorbidities, a better performance was observed for all case definitions in all weeks (aDOR 4.00; 95% CI 2.80–5.72 for WHO ILI, 1.91; 95% CI 1.44–2.52 for ECDC ILI and 1.77; 95% CI 1.34–2.34 for ECDC ARI) and in epidemic weeks (aDOR 3.14; 95% CI 1.92–5.10 for WHO ILI, 2.39; 95% CI 1.61–3.53 for ECDC ILI and 2.29; 95% CI 1.55–3.39 for ECDC ARI). Fever had the best performance in people aged ≥65 years (aDOR 6.82; 95% CI 4.36–10.65 for all weeks and 4.25; 95% CI 2.25–8.01 for epidemic weeks).

The performance of case definitions and symptoms in all weeks and in epidemic weeks for the influenza A and influenza B cases was similar (Figure 1).

For the influenza A virus, the WHO ILI case definition showed the best performance in both all weeks (aDOR 2.81; 95% CI 2.47–3.20) and in epidemic weeks (aDOR 2.16; 95% CI 1.82–2.57). The symptom that performed best in both was fever in all weeks (aDOR 4.29; 95% CI 3.46–5.32) and in epidemic weeks (2.72; 95% CI 2.06–3.59).

For the influenza B virus, the WHO ILI case definition also showed the best performance in both all weeks (aDOR 2.47; 95% CI 2.11–2.91) and in epidemic weeks (aDOR 2.23; 95% CI 1.81–2.75). Fever was the symptom with the best performance both in all weeks (aDOR 3.58, 95% CI 2.73–4.71) and in epidemic weeks (3.00, 95% CI 2.11–4.25).

## 4. Discussion

The importance of integrating data from sentinel sites with that from other medical and non-medical sources to detect and assess influenza epidemics has been pointed out [18,19,20]. However, each component of the integrated system must be assessed separately so as to improve data interpretation. This study assessed the performance of a primary healthcare sentinel surveillance system in detecting true influenza cases.

The proportion of laboratory-confirmed influenza cases obtained in all weeks was 31.3%, a value in the mid-range of the results of other studies, which range between 7.1% [21] and 52.2% [22]. The disparity of positive results reported by different authors [9,13,19,20,23,24] suggests that various specific circumstances influence the performance of each surveillance system.

The WHO and ECDC ILI case definitions for the surveillance of influenza in Catalonia performed differently, with differences in specific subgroups of patients, such as older people or those with comorbidities, although the results were similar when all weeks or only epidemic weeks were analyzed.

For the WHO and ECDC ILI case definitions, the overall sensitivities were 82% and 58%, and the specificities were 37% and 52%, respectively. A 2009–2014 French study [9] found higher sensitivities (89.8% for the WHO definition and 96.1% for the ECDC definition) but lower specificities (21.4% for the WHO definition and 6.6% for the ECDC definition). The differences may be because we only knew that patients had a temperature of >37.8 °C, instead of the required temperature of ≥38 °C stated in the WHO definition, and because in the French sentinel network, the ARI definition was an inclusion criterion, and nearly all patients included in the dataset had fever [25]. However, as in the French study, the WHO case definition of ILI was a better discriminator than the ECDC definition in our study.

The WHO ILI case definition had the best performance (aDOR 2.69; 95% CI 2.42–2.99) and was superior in people aged ≥65 years (aDOR 5.87; 95% CI 3.92–8.80) and in those with comorbidities (aDOR 4.00, 2.80–5.72). In contrast, the ILI definition of ECDC had a global performance of 1.52 (95% CI 1.40–1.66), and performed better in people aged ≥65 years (aDOR 2.32; 95% CI 1.62–3.34).

The ARI definition of ECDC had the worst overall performance (aDOR 1.45; 95% CI 1.33–1.58). This seems logical, as we compared confirmed cases of influenza against disease caused by other respiratory viruses or that were negative for influenza viruses, but not the performance of the definition in detecting any laboratory-confirmed influenza infection.

Another French study [26] during the 2012–2013 and 2013–2014 seasons concluded that the ILI ECDC definition performed poorly in tracking influenza epidemics, and suggested that the case definition adopted in any country should include fever. Our results support this idea, as the ARI case definition, which does not include fever, had the worst performance in confirmed influenza cases, both for all influenza A and B viruses jointly, and influenza A or B virus separately.

In a study carried out in Singapore between 2005 and 2009 in primary healthcare centers, the best performing ILI case definition was that of the WHO (DOR 13.5), while that of the ECDC was worse (DOR 9.7); the authors concluded that either definition is appropriate, while the ECDC ARI definition is inadequate for influenza surveillance [14].

Fever had the best performance in all weeks (aDOR 4.03; 95% CI 3.38–4.80) and in epidemic weeks (aDOR 2.78; 95% CI 2.21–3.50), and the best performance in both groups was in people aged ≥65 years. In contrast, a prospective study of children and adults in Canada during 2008–2011 found fever was discriminative for influenza infection in children, but not in adults [2].

Our results are in accordance with those of Falsey et al. [27] in the United States, which showed that fever was important in the elderly so as to retain specificity for the diagnosis of influenza.

Cough was the symptom with the second-best performance in all weeks (aDOR 1.85; 95% CI 1.65–2.08). Other studies have also found that the most predictive clinical manifestations are fever or cough. A review by Call et al. showed that, in spite of the different study periods considered, fever and cough were the most discriminating clinical manifestations, with a DOR ranging between 14 (95% CI 8.8–23.0) and 1.9 (95% CI 1.0–3.4) for fever, and between 12 (95% CI 1.4–97) and 1.4 (95% CI 0.71–5.0) for cough [13].

A United States study [28] found that measured or reported fever had a better capacity to discriminate the risk of true influenza infection than measured fever alone (DOR 1.95; 95% CI 1.73–2.20 and 1.79; 95% CI 1.59–2.0, respectively), with the performance of cough (DOR 6.99; 95% CI 5.60–8.73) being better than for cough and reported or measured fever (DOR 3.18; 95% CI 2.83–3.57).

In the study by Casalegno et al. [9], which was carried out in weeks 40 to 15, but not specifically in epidemic weeks, cough had the best performance (aDOR 2.53; 95% CI 2.23–290), with the best values in people aged ≥65 years (aDOR 5.55; 95% CI 2.67–11.52). In all weeks in our study, cough clearly performed worse than fever, with the best value in people aged 5–14 years rather than in older people.

In a 2009–2011 study in a Northern Indian rural community [21], the best performance was for cough (aDOR 3.1; 95% CI 1.5–6.7), followed by measured fever >38 °C (aDOR 2.5; 95% CI 1.3–4.9). In contrast, the reported fever was not associated with laboratory-confirmed influenza cases (aDOR 1.3; 95% CI 0.5–3.6).

A 2007–2008 United States study by Woolpert et al. [23] found that independent predictors of laboratory-confirmed influenza in the 6–49 years age group were cough and fever (aDOR 47.99; 95% CI 6.29–366.13 and 3.84; 95% CI 2.23–6.61, respectively), although acute symptom onset and, to a lesser extent, myalgia, were also predictors of laboratory-confirmed influenza. In our study, myalgia was a predictor in all age groups, but the sudden onset of symptoms was a poor predictor in adults. Unlike our results, where headache had the third best performance (aDOR 1.94; 95% CI 1.78–2.12), other authors found headache had the best performance (DOR 21.2; 95% CI 5.2–86.4) [14].

We found that sore throat was not a predictor of confirmed influenza in all weeks or in epidemic weeks, and was a discrete predictor only in the <5 years age group (aDOR 1.33; 95% CI 1.09–1.63). Likewise, it was not a significant predictor of confirmed influenza in all age groups in the United States study by Shah et al. [12] in 2009–2011, which found that the most discriminating clinical manifestations, with an aDOR >3, were cough and fever. Our results support the removal of sore throat from the clinical case definition in order to improve performance, as suggested by other authors [5].

A study by Boivin et al. in Canadian outpatient clinics in 1998–1999 concluded that cough and fever in a patient at a time when the influenza virus is circulating widely in the community is likely to be associated with influenza [29]. We found the discriminative capacity of the WHO ILI case definition that combines fever and cough was very close both in epidemic weeks and in all weeks (aDOR 2.20; 95% CI 1.90–2.54 and 2.69; 95% CI 2.42–2.99). In the review by Call et al. [13], the DOR for the combination of fever and cough ranged between 6.6 (95% CI 4.2–10) in patients aged ≥60 years and 3.6 (95% CI 3.1–4.2) in all ages.

No specific symptom can accurately diagnose influenza and, therefore, laboratory confirmation in representative samples of clinically compatible cases is crucial. This lack of specific influenza symptoms may explain, at least in part, why syndromic surveillance, which is routinely beneficial for the early detection of outbreaks of various infectious diseases, is not adequate for influenza surveillance [30].

As in the present study, Woolpert el al. [23] found no relevant differences in the clinical characteristics of influenza cases caused by the influenza A or B viruses, reinforcing the findings of other authors that influenza type A or B infection can affect different age or risk groups, but cause a similar clinical syndrome [31,32,33,34].

An interesting result of our study is that all case definitions assessed performed better in patients with comorbidities than in all patients. Most studies of the performance of the case definitions used for influenza surveillance do not distinguish between patients with or without comorbidities, and those that include comorbidities [21,29] do not analyze the performance of case definitions in patients with comorbidities separately, as we did. Future studies should consider this factor.

Our study had strengths and limitations. The main strength is that we only included patients with a laboratory-confirmed diagnosis of influenza, thus eliminating verification bias [13], and that all physicians were working in sentinel primary healthcare centers using the same criteria for clinical samples. Another strength is that other possible etiologies of acute respiratory infections were studied in all samples, excluding those with coinfection [24].

A first limitation is that, in our sentinel surveillance network, fever was predefined as a temperature >37.8 °C, but the specific temperature was not reported. In our opinion, the impact of this limitation may be minimal, as other factors, such as individual and daily variations, the site of measurement, and the natural trend for physicians to round temperatures up or down, can influence the measured temperature [9]. Secondly, age, the influenza type and subtype [5], or the season [18] might be partially responsible for differences in the performance of the variables studied. However, we adjusted for age and season, and it seems improbable that our results are invalid. Thirdly, during epidemic weeks, when physicians are aware of the onset of epidemic activity, the criteria for the sampling of suspected ILI patients might be less rigorous [28], but we have analyzed the performance of case definitions and clinical manifestations during both epidemic weeks and in all weeks, and no relevant differences were observed. Fourthly, although frequent in studies based on surveillance data, the number of patients aged ≥65 years was quite low (only 7.9% of all patients studied).

## 5. Conclusions

First, the WHO ILI case definition performed better than the ECDC ILI case definition, especially in older people. Secondly, all case definitions performed better in patients with comorbidities than in those without. Thirdly, the performance of the symptoms varied between age groups, with fever performing well in older people and cough in children, and in patients with comorbidities, fever performed best. However, no differences in the performance of case definitions or symptoms were found according to the influenza virus type.

## Figures and Tables

**Figure 1 viruses-12-00095-f001:**
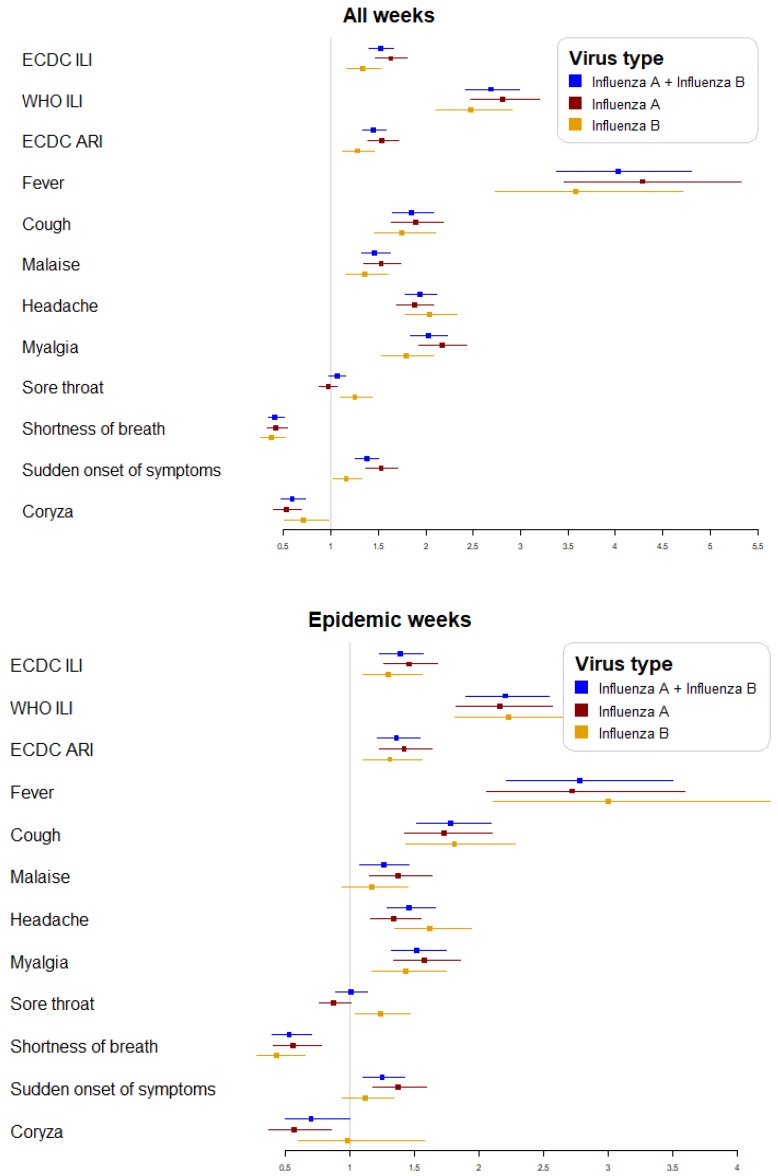
Adjusted DOR of case definitions and clinical symptoms for all weeks and epidemic weeks globally and according to influenza virus type.

**Table 1 viruses-12-00095-t001:** Influenza-positive and -negative patients included in the study. Influenza sentinel surveillance system, Catalonia, 2008–2018.

	All Cases N (%)	Positive for Influenza N (%)	Negative for Influenza N (%)
ALL WEEKS	10367	3241	7126
Case definition			
ECDC ILI	5309 (51.2%)	1892 (58.4%)	3417 (48.0%)
WHO ILI	7183 (69.3%)	2666 (82.3%)	4517 (63.4%)
ECDC ARI	5439 (52.5%)	1903 (58.7%)	3539 (49.7%)
Clinical symptoms			
Fever	8946 (86.3%)	3073 (94.8%)	5873 (82.4%)
Cough	8385 (80.9%)	2809 (86.7%)	5576 (78.2%)
Malaise	6290 (60.7%)	2167 (66.9%)	4123 (57.9%)
Headache	4242 (40.9%)	1650 (50.9%)	2592 (36.4%)
Myalgia	4785 (46.2%)	1778 (54.9%)	3007 (42.2%)
Sore throat	5158 (49.8%)	1643 (50.7%)	3515 (49.3%)
Shortness of breath	689 (6.6%)	114 (3.5%)	575 (8.1%)
Sudden onset of symptoms	6040 (58.3%)	2061 (63.6%)	3979 (55.8%)
Coryza	509 (4.9%)	108 (3.3%)	401 (5.6%)
Age			
0–4 years	2782 (26.8%)	593 (18.3%)	2189 (30.7%)
5–14 years	2580 (24.9%)	1145 (35.3%)	1435 (20.1%)
15–64 years	4188 (40.4%)	1341 (41.4%)	2847 (40.0%)
≥65 years	817 (7.9%)	162 (5.0%)	655 (9.2%)
Sex			
Female	5215 (50.3%)	1615 (49.8%)	3600 (50.5%)
Male	5150 (49.7%)	1626 (50.2%)	3524 (49.5%)
Comorbidities			
Yes	1132 (10.9%)	286 (8.8%)	846 (11.9%)
No	9235 (89.1%)	2955 (91.2%)	6280 (88.1%)

**Table 2 viruses-12-00095-t002:** Sensitivity, specificity, positive predictive value, likelihood ratios, crude and adjusted DOR of case definitions, and clinical symptoms for all weeks and for epidemic weeks. Influenza sentinel surveillance system, Catalonia, 2008–2018.

Case Definition	Se ^†^ (%)	Sp ^‡^ (%)	PPV ^§^ (%)	Positive LR ^¶^ (95% CI)	Negative LR (95% CI)	Crude DOR ^¥^ (95% CI)	Adjusted DOR (95% CI)
All weeks							
ECDC ILI	58	52	36	1.22 (1.17–1.26)	0.80 (0.76–0.84)	1.52 (1.40–1.66)	1.52 (1.40–1.66)
WHO ILI	82	37	37	1.30 (1.27–1.33)	0.48 (0.45–0.53)	2.68 (2.42–2.97)	2.69 (2.42–2.99)
ECDC ARI	59	50	35	1.18 (1.14–1.23)	0.82 (0.78–0.86)	1.44 (1.33–1.57)	1.45 (1.33–1.58)
Fever	95	18	34	1.15 (1.14–1.17)	0.29 (0.25–0.34)	3.90 (3.30–4.61)	4.03 (3.38–4.80)
Cough	87	22	34	1.11 (1.09–1.13)	0.61 (0.56–0.68)	1.81 (1.61–2.03)	1.85 (1.65–2.08)
Malaise	67	42	34	1.16 (1.12–1.19)	0.79 (0.74–0.83)	1.47 (1.35–1.60)	1.46 (1.32–1.63)
Headache	51	64	39	1.40 (1.34–1.47)	0.77 (0.74–0.80)	1.81 (1.67–1.97)	1.94 (1.78–2.12)
Myalgia	55	58	37	1.30 (1.24–1.36)	0.78 (0.75–0.82)	1.66 (1.53–1.81)	2.03 (1.84–2.23)
Sore throat	51	51	32	1.03 (0.99–1.07)	0.97 (0.93–1.01)	1.06 (0.97–1.15)	1.07 (0.98–1.16)
Shortness of breath	4	92	17	0.44 (0.36–0.53)	1.05 (1.04–1.06)	0.42 (0.34–0.51)	0.41 (0.34–0.51)
Sudden onset of symptoms	64	44	34	1.14 (1.10–1.18)	0.82 (0.78–0.87)	1.38 (1.27–1.51)	1.38 (1.26–1.50)
Coryza	3	94	21	0.59 (0.48–0.73)	1.02 (1.01–1.03)	0.58 (0.46–0.72)	0.59 (0.47–0.73)
Epidemic weeks							
ECDC ILI	58	50	60	1.16 (1.10–1.23)	0.84 (0.78–0.89)	1.39 (1.23–1.57)	1.39 (1.23–1.57)
WHO ILI	82	33	61	1.22 (1.18–1.27)	0.54 (0.49–0.60)	2.26 (1.96–2.60)	2.20 (1.90–2.54)
ECDC ARI	58	49	60	1.15 (1.09–1.22)	0.84 (0.79–0.90)	1.36 (1.21–1.54)	1.36 (1.21–1.54)
Fever	95	13	59	1.10 (1.07–1.12)	0.38 (0.30–0.46)	2.92 (2.33–3.66)	2.78 (2.21–3.50)
Cough	87	21	59	1.10 (1.07–1.13)	0.63 (0.55–0.72)	1.75 (1.49–2.06)	1.78 (1.52–2.09)
Malaise	72	31	57	1.04 (1.00–1.08)	0.91 (0.83–0.99)	1.15 (1.01–1.31)	1.26 (1.08–1.46)
Headache	51	57	61	1.20 (1.12–1.28)	0.85 (0.81–0.90)	1.40 (1.24–1.58)	1.46 (1.29–1.66)
Myalgia	57	49	59	1.11 (1.05–1.17)	0.89 (0.83–0.95)	1.25 (1.11–1.41)	1.52 (1.32–1.75)
Sore throat	51	49	56	1.00 (0.95–1.06)	1.00 (0.94–1.06)	1.01 (0.89–1.14)	1.01 (0.89–1.14)
Shortness of breath	4	93	39	0.50 (0.39–0.65)	1.04 (1.02–1.06)	0.48 (0.37–0.63)	0.53 (0.40–0.70)
Sudden onset of symptoms	63	42	58	1.09 (1.04–1.15)	0.87 (0.81–0.94)	1.25 (1.10–1.41)	1.25 (1.10–1.42)
Coryza	3	96	50	0.78 (0.56–1.07)	1.01 (1.00–1.02)	0.78 (0.55–1.08)	0.70 (0.50–1.00)

**^†^** Sensitivity; **^‡^** specificity; **^§^** positive predictive value; **^¶^** likelihood ratio; **^¥^** diagnostic odds ratio.

**Table 3 viruses-12-00095-t003:** Sensitivity, specificity, and adjusted DOR of clinical manifestations for all weeks, stratified by age group and comorbidities.

	0–4 Years			5–14 Years			15–64 Years			≥65 Years			Comorbidities			No Comorbidities		
	Se †	Sp ^‡^	Adjusted DOR ^§^ (95% CI)	Se ^†^	Sp ^‡^	Adjusted DOR ^§^ (95% CI)	Se ^†^	Sp ^‡^	Adjusted DOR ^§^ (95% CI)	Se ^†^	Sp ^‡^	Adjusted DOR ^§^ (95% CI)	Se ^†^	Sp ^‡^	Adjusted DOR ^§^ (95% CI)	Se ^†^	Sp ^‡^	Adjusted DOR ^§^ (95% CI)
Case definition																		
ECDC ILI	57	49	1.28 (1.06–1.54)	58	47	1.28 (1.09–1.50)	59	54	1.66 (1.45–1.90)	59	64	2.32 (1.62–3.34)	60	57	1.91 (1.44–2.52)	58	51	1.48 (1.35–1.62)
WHO ILI	86	19	1.47 (1.13–1.91)	85	32	2.86 (2.34–3.49)	79	46	3.16 (2.71–3.68)	75	64	5.87 (3.92–8.80)	85	41	4.00 (2.80–5.72)	82	36	2.61 (2.33–2.91)
ECDC ARI	58	47	1.23 (1.02–1.48)	58	47	1.26 (1.07–1.48)	60	53	1.61 (1.41–1.84)	59	59	1.97 (1.37–2.83)	60	55	1.77 (1.34–2.34)	59	50	1.41 (1.29–1.54)
Clinical symptoms																		
Fever	98	4	2.07 (1.09–3.91)	98	5	2.65 (1.63–4.32)	92	25	3.74 (3.01–4.66)	83	57	6.82 (4.36–10.65)	92	30	5.45 (3.43–8.66)	95	16	3.84 (3.18–4.64)
Cough	87	16	1.35 (1.03–1.79)	87	29	2.77 (2.25–3.42)	86	25	2.08 (1.74–2.48)	92	13	1.67 (0.89–3.13)	92	12	1.55 (0.97–2.50)	86	23	1.88 (1.66–2.12)
Malaise	57	56	1.60 (1.30–1.97)	59	45	1.18 (0.98–1.42)	77	31	1.33 (1.10–1.62)	80	36	2.16 (1.28–3.64)	87	25	2.32 (1.56–3.45)	65	45	1.40 (1.25–1.56)
Headache	21	92	2.93 (2.25–3.79)	53	55	1.38 (1.18–1.61)	62	46	1.40 (1.22–1.60)	50	64	1.82 (1.26–2.62)	55	61	2.03 (1.54–2.68)	50	64	1.93 (1.76–2.11)
Myalgia	17	93	2.48 (1.89–3.26)	36	69	1.23 (1.04–1.46)	85	28	2.16 (1.82–2.57)	81	45	3.25 (2.11–5.01)	69	49	2.66 (1.94–3.66)	54	59	1.96 (1.77–2.18)
Sore throat	30	76	1.33 (1.09–1.63)	59	39	0.90 (0.77–1.06)	53	38	0.70 (0.61–0.80)	50	48	0.89 (0.62–1.27)	47	51	0.92 (0.70–1.21)	51	51	1.08 (0.99–1.18)
Shortness of breath	2	89	0.18 (0.10–0.34)	2	95	0.49 (0.31–0.79)	4	94	0.74 (0.54–1.01)	12	83	0.52 (0.30–0.89)	15	79	0.66 (0.45–0.95)	2	94	0.33 (0.26–0.43)
Sudden onset of symptoms	63	41	1.21 (1.00–1.47)	62	40	1.14 (0.97–1.34)	65	46	1.54 (1.34–1.77)	62	55	1.88 (1.30–2.70)	64	51	1.80 (1.35–2.38)	64	43	1.34 (1.22–1.46)
Coryza	9	89	1.09 (0.80–1.48)	3	97	1.01 (0.65–1.56)	1	97	0.46 (0.26–0.81)	0	96	-	2	97	0.48 (0.18–1.26)	3	94	0.61 (0.48–0.76)

**^†^** Sensitivity; **^‡^** specificity; **^§^** diagnostic odds ratio.

**Table 4 viruses-12-00095-t004:** Sensitivity, specificity, and adjusted DOR of clinical manifestations for epidemic weeks, stratified by age group and comorbidities.

	0–4 Years			5–14 Years			15–64 Years			≥65 Years			Comorbidities			No Comorbidities		
	Se ^†^	Sp ^‡^	Adjusted DOR ^§^ (95% CI)	Se ^†^	Sp ^‡^	Adjusted DOR ^§^ (95% CI)	Se ^†^	Sp ^‡^	Adjusted DOR ^§^ (95% CI)	Se ^†^	Sp ^‡^	Adjusted DOR ^§^ (95% CI)	Se ^†^	Sp ^‡^	Adjusted DOR ^§^ (95% CI)	Se ^†^	Sp ^‡^	Adjusted DOR ^§^ (95% CI)
Case definition																		
ECDC ILI	58	47	1.29 (0.98–1.70)	57	49	1.29 (1.00–1.66)	59	50	1.41 (1.17–1.69)	59	62	2.34 (1.38–3.95)	61	61	2.39 (1.61–3.53)	58	49	1.30 (1.14–1.48)
WHO ILI	86	21	1.67 (1.16–2.40)	86	27	2.26 (1.67–3.06)	79	38	2.33 (1.90–2.86)	77	51	3.45 (1.95–6.09)	86	33	3.14 (1.92–5.10)	82	33	2.15 (1.85–2.50)
ECDC ARI	59	45	1.27 (0.96–1.67)	58	49	1.31 (1.01–1.68)	59	50	1.40 (1.17–1.69)	59	60	2.13 (1.26–3.60)	62	59	2.29 (1.55–3.39)	58	48	1.29 (1.13–1.46)
Clinical symptoms																		
Fever	98	4	1.57 (0.69–3.59)	98	4	2.52 (1.21–5.26)	92	17	2.55 (1.91–3.41)	85	43	4.25 (2.25–8.01)	92	19	2.70 (1.44–5.05)	95	12	2.80 (2.17–3.60)
Cough	88	17	1.54 (1.05–2.27)	87	25	2.18 (1.60–2.99)	85	24	1.87 (1.48–2.36)	92	12	1.60 (0.67–3.79)	93	14	2.42 (1.27–4.61)	86	22	1.75 (1.48–2.07)
Malaise	61	49	1.51 (1.13–2.03)	65	33	1.03 (0.77–1.37)	81	21	1.20 (0.92–1.57)	85	25	1.32 (0.60–2.90)	89	17	1.92 (1.08–3.43)	70	33	1.21 (1.03–1.41)
Headache	22	89	2.25 (1.55–3.27)	54	54	1.34 (1.06–1.71)	61	42	1.13 (0.94–1.36)	50	61	1.54 (0.90–2.63)	54	61	1.96 (1.32–2.92)	51	57	1.42 (1.24–1.62)
Myalgia	18	90	1.79 (1.21–2.66)	38	65	1.19 (0.91–1.54)	86	23	1.81 (1.42–2.29)	82	35	2.07 (1.13–3.80)	69	41	2.01 (1.29–3.12)	55	50	1.47 (1.27–1.71)
Sore throat	33	74	1.33 (0.99–1.77)	60	33	0.77 (0.59–0.99)	53	42	0.82 (0.68–0.99)	48	51	0.99 (0.59–1.66)	47	50	0.88 (0.60–1.28)	52	49	1.02 (0.90–1.16)
Shortness of breath	2	92	0.22 (0.10–0.52)	3	96	0.79 (0.40–1.59)	5	93	0.66 (0.44–0.99)	12	83	0.53 (0.25–1.12)	16	80	0.75 (0.45–1.23)	2	94	0.41 (0.29–0.59)
Sudden onset of symptoms	63	38	1.16 (0.87–1.55)	61	43	1.19 (0.92–1.53)	65	42	1.30 (1.08–1.56)	62	55	1.93 (1.15–3.27)	65	53	2.01 (1.36–2.97)	63	41	1.18 (1.03–1.35)
Coryza	9	91	0.99 (0.63–1.57)	3	97	0.91 (0.43–1.93)	1	99	0.64 (0.25–1.63)	0	97	-	1	99	1.26 (0.19–8.11)	3	96	0.69 (0.49–0.98)
Fever and Cough	98	4	1.57 (0.69–3.59)	98	4	2.52 (1.21–5.26)	92	17	2.55 (1.91–3.41)	85	43	4.25 (2.25–8.01)	92	19	2.70 (1.44–5.05)	95	12	2.80 (2.17–3.60)
Fever and myalgia	88	17	1.54 (1.05–2.27)	87	25	2.18 (1.60–2.99)	85	24	1.87 (1.48–2.36)	92	12	1.60 (0.67–3.79)	93	14	2.42 (1.27–4.61)	86	22	1.75 (1.48–2.07)

**^†^** Sensitivity; **^‡^** specificity; **^§^** Diagnostic odds ratio.

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
