# Peer review of "Usefulness of Clinical Definitions of Influenza for Public Health Surveillance Purposes"

_viruses, 2020, doi:10.3390/v12010095_

Round 1

Reviewer 1 Report

Reviewer’s report

This manuscript describes the performance of two Influenza case definitions (WHO and ECDC) within a primary European healthcare surveillance system.

As the author states, we know that the performance of ILI definitions may change from a surveillance system (or population) to another. EU national case definition differences hampered the comparability of data at the European and international level. So I find it quite relevant to further investigate the performance of ILI case definition in different European population and surveillance system.

The authors mainly described and compared the performance (Se, Sp, predictive value, LR, diagnostic odds ratios) of WHO ILI, ECDC ILI and several symptoms (alone and in association) with the laboratory confirmed Influenza detection. The analysis was stratified by age group and epidemic season. The main result is that cough and fever are the best predictors for Flu.

General comments

The paper is nicely written, quite straight forward, and the results are clearly present. I also quite agree with the main results. Still I have two general comments:

First I feel that the authors should highlight more one key weakness of this kind of dataset and the potential impact of this on the performance observed.

What is the underlying selection criterion for influenza testing which generated this dataset? Is there any influenza definition (ARI or ILI) use in the network? What is the strategy to sample and test? Is there evidence of variability in the implementation of these criteria across the multiple study sites involved? A short description of the network (number of sentinel GP, number of sentinel laboratories, core lab ?, clinical definition use should be provided in the material and methods) The fever definition. Indeed, at the individual level the specific value of temperature is submitted to many variations factors. However at the dataset level it may impact the inclusion criteria, the clinical case definition as defined in the study dataset, and in fine the observed performance. I agree with the discussion part but : The definition of fever (>37.8) must be present in the inclusion criteria (and not just in the discussion part) in the material and methods part It should be added that the WHO definition (>= 38) was tested with this level of fever

Second. In this work, I don’t understand the difference between the WHO ILI definition (fever and cough and onset within the last 10 days) and the association of Fever and Cough. If there is only one level of temperature defined, it applied to both. Are there some patients in your dataset test after 10 days of onset?? In any case the performances are almost identical. I would suggest to clearly state that you tested the WHO definition with a fever level defined at 37.8 degree and only present the WHO definition performance. Moreover this work is not design to generate and tests new ILI definitions. In that case all the other possible symptom associations (Fever AND/OR COUGH AND/OR MALAISE AND/OR…) should be tested and compared (not just fever and cough, fever and myalgia). I would not present the performance of these two clinical associations alone because there are many questions pending on the methodology/strategy used by the authors to selected only those two and the guarantee that there are not more optimal influenza clinical definition that could be generate with that dataset. I would substitute the WHO definition to Fever and cough. The main message would be much more relevant and the tables clearer.

Specific comments

Introduction line 44: “affects 10-20% of the unvaccinated population” please provide a reference for that quote Introduction line 62: “more recent studies” there is only one reference and the study was published in 2003. Please provide more recent references Results table 1 : Results for the positive and negative results for influenza in the different age group are duplicated in the table (ex: 0-4 years 452 X 2) Discussion line 15: In this study the sensitivities were probably higher because the GROG ARI definition was an inclusion criterion and that nearly all the patients included in the dataset did had fever (but it is not an ILI definition). Indeed what matter the most is that the WHO definition had the best performance. It also probably why cough was (slightly) more predictive for flu than fever compare with this study (discussion line 54) Discussion line 94: In the limitation I would add that the number of patients age >65 years is quite low (only 7.9% of the whole dataset). It is quite usual for any influenza surveillance system but it have to be acknowledged that some conclusions are drawn on a small part of this dataset.

Author Response

Dear Editor and reviewer 1, 

Please find enclosed the revised version of the manuscript entitled "Usefulness of clinical definitions of influenza for public health surveillance purposes". The manuscript has been revised taking into account the reviewers’ comments, as requested. A detailed response to the reviewers’ comments is enclosed.

Sincerely yours,

Núria Soldevila

Reviewer 1

Reviewer’s report

This manuscript describes the performance of two Influenza case definitions (WHO and ECDC) within a primary European healthcare surveillance system. As the author states, we know that the performance of ILI definitions may change from a surveillance system (or population) to another. EU national case definition differences hampered the comparability of data at the European and international level. So I find it quite relevant to further investigate the performance of ILI case definition in different European population and surveillance system.

The authors mainly described and compared the performance (Se, Sp, predictive value, LR, diagnostic odds ratios) of WHO ILI, ECDC ILI and several symptoms (alone and in association) with the laboratory confirmed Influenza detection. The analysis was stratified by age group and epidemic season. The main result is that cough and fever are the best predictors for Flu.

General comments

The paper is nicely written, quite straight forward, and the results are clearly present. I also quite agree with the main results. Still I have two general comments:

First I feel that the authors should highlight more one key weakness of this kind of dataset and the potential impact of this on the performance observed. What is the underlying selection criterion for influenza testing which generated this dataset? Is there any influenza definition (ARI or ILI) use in the network?

There was a clinical presentation criterion underlying the selection for testing a patient: individuals who presented symptoms compatible with acute respiratory infection (without any specified case definition) or ILI according to the European ILI case definition (https://eur-lex.europa.eu/legal-content/EN/TXT/PDF/?uri=CELEX:32018D0945&from=EN#page=24).

What is the strategy to sample and test?

Simply a random sampling of up to two patients attended by the sentinel physician who presented symptoms compatible with ARI o ILI according to the European ILI case definition. This information has been added to the Study design section in the revised manuscript.

Is there evidence of variability in the implementation of these criteria across the multiple study sites involved?

No there was no variability in the implementation of the sampling selection criteria. Sentinel physicians participating in the surveillance remain mostly the same, season after season. There is little turnover and, when new physicians are recruited, they receive personalized training in the surveillance system’s guidelines from the public health professionals in charge.

A short description of the network (number of sentinel GP, number of sentinel laboratories, core lab ?, clinical definition use should be provided in the material and methods)

A summarized description of the network is provided in the methods section, as well as the clinical definition stated above.

The fever definition. Indeed, at the individual level the specific value of temperature is submitted to many variations factors. However at the dataset level it may impact the inclusion criteria, the clinical case definition as defined in the study dataset, and in fine the observed performance. I agree with the discussion part but : The definition of fever (>37.8) must be present in the inclusion criteria (and not just in the discussion part) in the material and methods part It should be added that the WHO definition (>= 38) was tested with this level of fever.

Fever was defined as temperature >37.8oC and this has been included in the Data collection section and also as a limitation of the study.

Second. In this work, I don’t understand the difference between the WHO ILI definition (fever and cough and onset within the last 10 days) and the association of Fever and Cough. If there is only one level of temperature defined, it applied to both. Are there some patients in your dataset test after 10 days of onset?? In any case the performances are almost identical. I would suggest to clearly state that you tested the WHO definition with a fever level defined at 37.8 degree and only present the WHO definition performance. Moreover this work is not design to generate and tests new ILI definitions. In that case all the other possible symptom associations (Fever AND/OR COUGH AND/OR MALAISE AND/OR…) should be tested and compared (not just fever and cough, fever and myalgia). I would not present the performance of these two clinical associations alone because there are many questions pending on the methodology/strategy used by the authors to selected only those two and the guarantee that there are not more optimal influenza clinical definition that could be generate with that dataset. I would substitute the WHO definition to Fever and cough. The main message would be much more relevant and the tables clearer.

We agree. We have deleted the combinations “fever and cough” and “fever and myalgia” from the results and from tables 2, 3 and 4. In the Discussion section of the revised manuscript we compare the performance of the WHO ILI case definition, which includes fever and cough, with the combinations of fever and cough studied by other authors.

In addition, in Table 1 we have deleted the information related to epidemic weeks because the characteristics of the patients studied is well represented with the description related to all weeks.

Specific comments

Introduction line 44: “affects 10-20% of the unvaccinated population” please provide a reference for that quote

We have provided a reference for this quote.

Introduction line 62: “more recent studies” there is only one reference and the study was published in 2003. Please provide more recent references

We have introduced three references corresponding to studies that use the Diagnostic Odds Ratio as a measure of the performance.

Results table 1: Results for the positive and negative results for influenza in the different age group are duplicated in the table (ex: 0-4 years 452 X 2)

The reviewer is correct. We have corrected this in the revised manuscript.

Discussion line 15: In this study the sensitivities were probably higher because the GROG ARI definition was an inclusion criterion and that nearly all the patients included in the dataset did had fever (but it is not an ILI definition). Indeed what matter the most is that the WHO definition had the best performance. It also probably why cough was (slightly) more predictive for flu than fever compare with this study (discussion line 54).

We agree. We have changed the explanation of the differences observed in the present study and in the French study.

Discussion line 94: In the limitation I would add that the number of patients age >65 years is quite low (only 7.9% of the whole dataset). It is quite usual for any influenza surveillance system but it have to be acknowledged that some conclusions are drawn on a small part of this dataset.

We agree. This limitation has been included in the revised manuscript.

* The attached document shows the revisions for both reviewers.

Reviewer 2 Report

I only have minor comments:

It would be useful that the authors define what they intend by all weeks. Does all weeks mean from week 1-52/3 (what I assumed) or a specific surveillance period? The diagnostic test detects influenza A-C, so were influenza positive C cases also included in this study (if any detected). Some studies treating of influenza case definition deliberately exclude patients’ positive test for influenza C as it is considered to make only mild infections clinically distinct from A and B. Could the authors comment on this point? Did the authors see differences the case definition performance in-between influenza A, B and eventually C (if any detected). Maybe their test does not allow discriminating different influenza types. If it is the case, this should ideally be specified as several diagnostic influenza tests detect Influenza A and B.

There is a lot of data leading to crowded tables not always easy to read, maybe adding some forest plots would have been a good more “visual” option.

Author Response

Dear reviewer 2, 

Please find enclosed the revised version of the manuscript entitled "Usefulness of clinical definitions of influenza for public health surveillance purposes". The manuscript has been revised taking into account the reviewers’ comments, as requested. A detailed response to the reviewers’ comments is enclosed.

Sincerely yours,

Núria Soldevila

Reviewer 2

I only have minor comments:

It would be useful that the authors define what they intend by all weeks. Does all weeks mean from week 1-52/3 (what I assumed) or a specific surveillance period?

We now state in the revised version of the manuscript that “all weeks” refers to weeks 1 to 52/53 and that epidemic weeks refer to weeks with an ILI incidence rate above the set epidemic threshold per each season.

The diagnostic test detects influenza A-C, so were influenza positive C cases also included in this study (if any detected).

Some studies treating of influenza case definition deliberately exclude patients’ positive test for influenza C as it is considered to make only mild infections clinically distinct from A and B. Could the authors comment on this point?

The reviewer is correct. There were 25 cases of influenza C virus during the study period and, because virus C does not cause epidemics, cases of influenza C have been deleted in the revised version of the manuscript.

Did the authors see differences the case definition performance in-between influenza A, B and eventually C (if any detected). Maybe their test does not allow discriminating different influenza types. If it is the case, this should ideally be specified as several diagnostic influenza tests detect Influenza A and B. There is a lot of data leading to crowded tables not always easy to read, maybe adding some forest plots would have been a good more “visual” option.

We have added a figure showing the performance of the different case definitions and symptoms assessed considering influenza A confirmed cases or influenza B confirmed cases as positive. We have tried to put this information in a forest plot instead of a table, as suggested.

However, for the results shown in tables 2, 3 and 4 we consider that tables are a better option than forest plots because, in this way, readers have specific figures on the performance of case definitions and symptoms and can better compare them with the results of other studies.

* The attached document shows the revisions for both reviewers.
